# Short-Term Outcomes of Conventional Laparoscopic versus Robot-Assisted Distal Pancreatectomy for Malignancy: Evidence from US National Inpatient Sample, 2005–2018

**DOI:** 10.3390/cancers16051003

**Published:** 2024-02-29

**Authors:** Jyun-Ming Huang, Sheng-Hsien Chen, Te-Hung Chen

**Affiliations:** 1Department of Surgery, China Medical University Hospital, No. 2, Yude Rd., North Dist., Taichung City 404327, Taiwan; 2School of Medicine, China Medical University, No. 2, Yude Rd., North Dist., Taichung City 404, Taiwan

**Keywords:** pancreatic cancer, perioperative complications, robot-assisted surgery, surgical outcomes

## Abstract

**Simple Summary:**

This study investigated the short-term outcomes of laparoscopic versus robot-assisted distal pancreatectomy for pancreatic cancer using the US Nationwide Inpatient Sample (NIS) database from 2005 to 2018. Data from 886 patients were analyzed. Despite higher costs, robot-assisted surgery was associated with fewer complications, reduced risk of blood transfusion, and a shorter hospital stay compared to conventional laparoscopy. The findings suggest that while robot-assisted surgery comes with increased expenses, it may offer better short-term outcomes for patients with pancreatic cancer.

**Abstract:**

Background: The primary treatment for pancreatic cancer is surgical resection, and laparoscopic resection offers benefits over open surgery. This study aimed to compare the short-term outcomes of robot-assisted vs. conventional laparoscopic distal pancreatectomy. Methods: Data of adults ≥ 20 years old with pancreatic cancer who underwent conventional laparoscopic or robot-assisted laparoscopic distal pancreatectomy were extracted from the United States (US) Nationwide Inpatient Sample (NIS) 2005–2018 database. Comorbidities and complications were identified through the International Classification of Diseases (ICD) codes. Short-term outcomes were compared using logistic regression and included length of hospital stay (LOS), perioperative complications, in-hospital mortality, unfavorable discharge, and total hospital costs. Results: A total of 886 patients were included; 27% received robot-assisted, and 73% received conventional laparoscopic surgery. The mean age of all patients was 65.3 years, and 52% were females. Multivariable analysis revealed that robot-assisted surgery was associated with a significantly reduced risk of perioperative complications (adjusted odds ratio (aOR) = 0.61, 95% confidence interval (CI): 0.45–0.83) compared to conventional laparoscopic surgery. Specifically, robot-assisted surgery was associated with a significantly decreased risk of VTE (aOR = 0.35, 95% CI: 0.14–0.83) and postoperative blood transfusion (aOR = 0.37, 95% CI: 0.23–0.61). Robot-assisted surgery was associated with a significantly shorter LOS (0.76 days shorter, 95% CI: −1.43–−0.09) but greater total hospital costs (18,284 USD greater, 95% CI: 4369.03–32,200.70) than conventional laparoscopic surgery. Conclusions: Despite the higher costs, robot-assisted distal pancreatectomy is associated with decreased risk of complications and shorter hospital stays than conventional laparoscopic distal pancreatectomy.

## 1. Introduction

Pancreatic cancer is the seventh principal cause of cancer-related mortality in industrialized nations [1,2]. Notably, within the United States (US) it is the third most prevalent form of cancer [3]. Pancreatic cancer is also associated with extremely high mortality; the 5-year survival rate for persons who undergo surgery is <17%, and for those not treated with surgery is <1% [4].

Surgical excision is the only intervention with the potential for increasing survival time, but in many patients, the disease is diagnosed at a late stage and thus surgery is confined to a small subset of patients. There are several surgical interventions, including pancreatoduodenectomy (commonly known as the Whipple procedure), pylorus-preserving pancreaticoduodenectomy, and distal pancreatectomy [5,6]. Distal pancreatectomy is a procedure that is used to manage benign and malignant lesions in the body or tail of the pancreas [7,8]. This operation entails the removal of the left-of-midline portion of the pancreas, with the duodenum and distal bile duct remaining outside the excised area [8].

Conventional distal pancreatectomy is performed via an open surgical approach; however, with advances in technology distal pancreatectomy can be performed with laparoscopic techniques, and most recently, robot-assisted laparoscopic distal pancreatectomy has been developed [9,10]. Prior investigations have indicated that laparoscopic distal pancreatectomy offers distinct advantages over open resection, including decreased postoperative pain, reduced risks of complications, and shorter hospital stays [11,12]. Laparoscopic distal pancreatectomy is becoming the procedure of choice; however, robot-assisted distal pancreatectomy may have advantages over open and laparoscopic approaches. 

Thus, the purpose of this study was to compare the short-term outcomes of laparoscopic and robot-assisted distal pancreatectomy for patients with pancreatic cancer using information from a nationwide database.

## 2. Patients and Methods

### 2.1. Data Source

Data for this study were extracted from the 2005–2018 Nationwide Inpatient Sample (NIS), a database developed by the Healthcare Cost and Utilization Project (HCUP) in the US [13]. The NIS is maintained by the Agency for Healthcare Research and Quality (AHRQ) [13]. The NIS database represents a 20% sample of inpatient admissions from 45 states and 1051 hospitals in the US. Principal and secondary diagnoses, principal and secondary procedures, admission date and diagnosis, discharge status, patient demographic data, and length of hospital stay (LOS) are included for each inpatient. Statistical weights that allow generalized estimates of national case volumes are also provided in the NIS. 

### 2.2. Study Design and Approvals

Data extracted from the 2005–2018 NIS were retrospectively reviewed. Permission to use the database was obtained from HCUP-NIS (certificate number-HCUP-349J94IVU), and the study conforms to the data-use agreement for the NIS database from HCUP [13]. Because this study analyzed data collected by the NIS and did not directly involve patients, and patient data in the NIS database are deidentified, signed informed consent was waived for this present study. 

### 2.3. Study Population 

Patients ≥ 20 years old with primary pancreatic cancer were identified in the NIS database based on the International Classification of Diseases, Ninth and Tenth revisions, and Clinical Modification codes (ICD-9-CM and ICD-10-CM). Patients who underwent conventional laparoscopic or robot-assisted laparoscopic distal pancreatectomies were further identified by ICD procedure codes (ICD-9-PCS and ICD-10-PCS). Patients with malignant neoplasms located at the head of the pancreas and those who had undergone the Whipple procedure were excluded from the analysis. In addition, patients without complete information on sex and main study outcomes were also excluded.

### 2.4. Main Outcomes and Study Variables

The outcomes of this study were in-hospital mortality, unfavorable discharge (defined as discharge to long-term care facilities), LOS, total hospital costs, and the occurrence of perioperative complications. Complications considered were acute myocardial infarction (AMI), cerebrovascular accident (CVA), venous thromboembolism (VTE), periprocedural shock, hypertension, other cardiovascular complications, pneumonia, postprocedural pneumothorax, postprocedural air leak, acute respiratory failure, pulmonary collapse (atelectasis), sepsis, infection, mechanical ventilation, postoperative blood transfusion, other complications of the respiratory system, and perforations of organ or vessels. Complications were identified by ICD codes.

Data extracted from the medical records included patient demographic information (age and sex), insurance status, household income, and year of admission (categorized into admissions during 2005–2014 and 2015–2018). Smoking status, comorbidities, Charlson Comorbidity Index (CCI), whether or not the spleen was removed, weekend admission, emergent admission, and preoperative characteristics were also identified and extracted from the database. Comorbidities identified and included in the analysis were coronary artery disease (CAD), congestive heart failure (CHF), persistent anemia, diabetes, hypertension, cerebrovascular disease, chronic pulmonary disease, hyperlipidemia, drug abuse, severe liver disease, moderate or severe renal disease, and rheumatic disease. Hospital characteristics examined were hospital bed number and hospital location/teaching status.

The ICD codes used in the analyses are summarized in Appendix A.

### 2.5. Statistical Analysis

Descriptive statistics of the patients were presented as unweighted counts (n) and weighted percentage (%), or mean ± standard error (SE). Since the NIS database covers a 20% sample of the US annual inpatient admissions, weighted samples (before 2011 using TrendWT and after 2012 using DISCWT), stratum (NIS_STRATUM), and cluster (HOSPID) were used to produce national estimates for all analyses. PROC SURVEYFREQ and SURVEYREG procedures were used for comparing categorical and continuous data between the groups, respectively. Logistic regression models were performed with PROC SURVEYLOGISTIC and SURVEYREG to identify the risk for in-hospital mortality, unfavorable discharge, any perioperative complication, LOS, and hospital cost of patients who received the 2 types of surgery. Multivariable regression was adjusted for related variables with a value of *p* < 0.05 in Table 1, including age (continuous), smoking, hospital location/teaching status, emergency admission, and CAD. All *p* values were 2-sided, and a value of *p* < 0.05 was considered statistically significant. All statistical analyses were performed using the statistical software package SAS software version 9.4 (SAS Institute Inc., Cary, NC, USA).

## 3. Results

### 3.1. Patient Selection

The patient selection process is summarized in Figure 1. A total of 1209 patients ≥ 20 years old with a diagnosis of malignant neoplasm of the pancreas who underwent laparoscopic or robot-assisted laparoscopic distal pancreatectomy were identified in the NIS database. Patients with missing data on sex and main outcomes (*n* = 17) were excluded. We further excluded 241 patients with malignant neoplasm of the head of the pancreas and 65 patients who received a proximal pancreatectomy. Finally, 886 patients were included in the analysis. This sample can be extrapolated to 4397 individuals in the entire US after applying the sample weights provided by the dataset (Figure 1).

### 3.2. Patient Characteristics 

Patient information, including demographic data, comorbidities, surgery type, hospital-related information, and in-hospital outcomes, are summarized in Table 1. Of the patients, 27% received robot-assisted distal pancreatectomy and 73% received conventional laparoscopic distal pancreatectomy. The mean age of all patients was 65.3 years, 465 patients (52.5%) were females, 306 patients (34.6%) were smokers, and 80 patients (9.1%) were admitted emergently. The most common comorbidity was hypertension (52.0%). 

### 3.3. In-Hospital Outcomes

In-hospital outcomes of the patients are summarized in Table 2. The group of patients that received robot-assisted distal pancreatectomy had significantly lower proportions of any perioperative complication (18.4% vs. 25.8%), VTE (1.6% vs. 4.5%), and postoperative blood transfusion (3.4% vs. 8.0%) compared to those that received conventional laparoscopic surgery (all, *p* < 0.05). In addition, the robot-assisted surgery group had a shorter mean LOS than the conventional laparoscopic surgery group (6.0 vs. 6.8 days). However, the total hospital costs were significantly higher in the robot-assisted group (112,036 vs. 95,402 USD) (Table 2).

### 3.4. Associations between Type of Surgery, In-Hospital Outcomes, and Costs 

Associations between type of surgery and in-hospital outcomes are summarized in Table 3 and Table 4. After adjusting for relevant confounders in the multivariable analysis, robot-assisted surgery was associated with a significantly decreased risk of any perioperative complication (adjusted odds ratio (aOR) = 0.61, 95% confidence interval (CI): 0.45–0.83, *p* = 0.002) as compared with conventional laparoscopy. Concerning specific perioperative complications, robot-assisted surgery was associated with a significantly decreased risk VTE (aOR = 0.35, 95% CI: 0.14–0.83, *p* = 0.021) and postoperative blood transfusion (aOR = 0.37, 95% CI: 0.23–0.61, *p* < 0.001, respectively) (Table 3). 

Multivariable analysis showed robot-assisted surgery resulted in a 0.76-day shorter length of stay (LOS) compared to laparoscopic surgery (95% CI: −1.43–−0.09) and incurred about 18,000 USD higher total hospital costs (95% CI: 4369–32,200) than conventional laparoscopic surgery. (Table 4).

## 4. Discussion

As compared to laparoscopic distal pancreatectomy for pancreatic cancer, robot-assisted surgery was independently associated with an approximately 65% reduction in VTE risk and a 63% reduction in postoperative blood transfusions. Robot-assisted surgery was associated with a shorter LOS but a greater total hospital cost. The findings indicate that robot-assisted distal pancreatectomy provides advantages in terms of some short-term outcomes over conventional laparoscopic surgery, despite the higher cost. 

Over the past decade, robot-assisted surgical procedures have been developed in almost every surgical field [14,15]. Laparoscopic surgery has clear advantages over open surgery, including less postoperative pain and faster recovery [16]. Robot-assisted procedures have these same advantages and may offer better results in technically demanding procedures [14]. Robot-assisted pancreatic surgery is becoming more common, and while the learning curve is steep, familiarity with robotic surgery in general shortens the learning curves for different procedures [17,18,19].

Over the past 6 to 7 years, several studies have compared laparoscopic and robot-assisted distal pancreatectomy and have provided results similar to those of our study; overall, robotic-assisted surgery offers at last equivalent results as laparoscopic surgery, but with a higher overall medical cost [20,21,22,23,24,25,26]. The higher cost has led to some questioning the overall feasibility of a robot-assisted approach [21,22,26]. 

Our study did not compare the resection margin, lymph node yields, or conversion rates following laparoscopic versus robot-assisted surgery due to a lack of reliable information in the dataset. A recent retrospective cohort study found that the R0 resection rate was similar between laparoscopic and robot-assisted distal pancreatectomy (76% vs. 70%, *p* = 0.04), as was 90-day mortality and overall survival [20]. Notably, the lymph node yield was higher with the robot-assisted procedure and the conversion rate to an open procedure was lower; however, laparoscopic surgery was associated with fewer major complications. A propensity score-matched analysis reported that compared to laparoscopic distal pancreatectomy, the robot approach was associated with a significantly higher rate of R0 resection (91% vs. 62%, *p* = 0.042) and lower open conversion rate (0% vs. 5%, *p* < 0.001) [25]. The median disease-free survival (DFS) and overall survival (OS) were similar between the two procedures; however, robotic surgery was associated with a significantly longer operation time and cost. 

In our results, the proportion of combined spleen removal was not different between the two approaches. A multi-center analysis that matched patients undergoing laparoscopic and robot-assisted distal pancreatectomy (402 patients in each group) reported that the robot-assisted procedure was associated with a longer operation time, lower conversion rate, higher spleen preservation rate, longer hospital stay, and lower readmission rate (all, *p* < 0.05) [24]. Jiang et al. [23] also reported that robot-assisted distal pancreatectomy was associated with a significantly higher spleen preservation rate compared to laparoscopic surgery in patients with benign and low-grade malignant lesions.

As robot-assisted distal pancreatectomy has become a more commonly performed procedure, some meta-analyses have been performed comparing the robot-assisted and laparoscopic approaches [27,28,29,30]. Three of the meta-analyses were published in 2020. Zhou et al. [30] included seven studies from high-volume robotic surgery centers with a total of 2264 patients. The authors reported that robot-assisted distal pancreatectomy was associated with lower estimated blood loss, lower blood transfusion rate, lower postoperative mortality rate, and shorter length of hospital stay. There was no significant difference between the two procedures in operation time, the number of lymph nodes harvested, positive margin rate, spleen preservation rate, rate of severe morbidity, incidence of postoperative pancreatic fistula, and severe postoperative pancreatic fistula (grade B and C). A network meta-analysis by Lyu et al. [28] included 46 trials with 8377 patients, and the surface under the cumulative ranking curve (SUCRA) was used to determine the probability of which method would be the best for each outcome measure examined in the study. Robot-assisted surgery had the highest probability of having the smallest estimated blood loss (SUCRA = 91%), the lowest incidence of postoperative pancreatic fistula (SUCRA = 95%), postoperative bleeding (SUCRA = 75%), overall complications (SUCRA = 879%) and major complications (SUCRA = 99%), lowest mortality rate (SUCRA = 83), and highest probability of attaining as R0 resection (SUCRA = 75%). A meta-analysis by Hu et al. [27] included four prospective studies and 18 retrospective studies, and found that the spleen preservation rate was higher with robot-assisted distal pancreatectomy; but there were no differences in the R0 resection rate, mortality rate, number of lymph nodes harvested, and complication and pancreatic fistula rates between laparoscopic and robot-assisted surgery.

The most recent meta-analysis was published in 2023 by van Ramshorst et al. [29]. The analysis included 43 studies with 6757 patients. Consistent with prior studies, the robot-assisted procedure was associated with higher cost and longer operation time, but significantly less blood loss, a lower conversion rate, and a lower rate of unplanned splenectomies. In the group of patients with pancreatic ductal adenocarcinoma, the robot-assisted approach had a higher lymph node yield but a similar R0 resection rate compared to laparoscopic surgery. 

### Strengths and Limitations

A large-scale and nationally representative database was used in this study, and thus the results and conclusions of this study can be generalized to the whole population. Nevertheless, this study has several limitations. Firstly, detailed characteristics of the tumor, such as tumor size, staging, grade, anatomic features, and invading vasculatures, could not be studied as these data are not recorded in the NIS database. Similarly, medications prescribed could not be considered in the analysis. Secondly, the study relied on the accuracy of the ICD coding system, where coding errors might exist. Thirdly, perioperative care tends to improve over time, with later years theoretically offering better care, leading to fewer complications. Given that the robotic approach is relatively newer, it may have been more frequently utilized in the later years of our study period, potentially introducing a source of bias. In addition, the specific criteria for selecting the robotic approach remain unclear and cannot be considered in the analysis due to the NIS’s reliance on claim codes for data collection. To address these concerns, we incorporated the year of admission as a variable in our analysis and discovered that its distribution was balanced between the two surgical approaches. Nevertheless, we advise readers to interpret the outcomes with caution, considering this context. Fourthly, other potential confounders such as preoperative performance status, clinical laboratory measures, or intraoperative parameters such as operation time are not collected in the NIS database, as well as resection margin and lymph node yields. Fifth, although conversion rate is crucial when assessing the outcomes between laparoscopic and robotic approaches, however, it cannot be well defined using the claim codes and thus could not be included in the analyses. Furthermore, robot-assisted surgery is typically carried out by hospitals and surgeons with a high level of experience. However, the absence of data on surgeons’ experience in the dataset precluded further analysis. Lastly, since the NIS does not contain follow-up data after discharge, mid- and long-term oncological outcomes could not be compared.

## 5. Conclusions

Despite its higher cost compared to conventional laparoscopic surgery, robot-assisted distal pancreatectomy has demonstrated superior in-hospital outcomes and a reduced risk of perioperative complications. This study’s results underscore important factors to consider when choosing between robotic and conventional laparoscopic surgery.

## Figures and Tables

**Figure 1 cancers-16-01003-f001:**
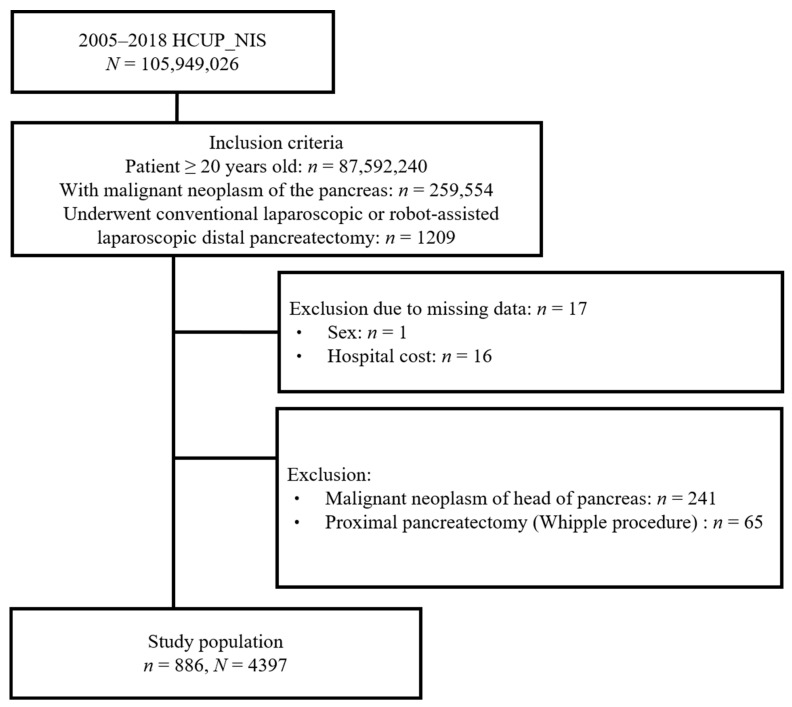
Flow diagram of patient selection and inclusion.

**Table 1 cancers-16-01003-t001:** Patient characteristics.

Characteristics	Total (*n* = 886)	Laparoscopic Surgery		*p*-Value
		Conventional (*n* = 648)	Robot-Assisted (*n* = 238)	
Age, years	65.3 ± 0.4	64.7 ± 0.4	66.8 ± 0.6	**0.040**
20–29	17 (2.0)	13 (2.1)	4 (1.7)	0.058
30–39	23 (2.6)	16 (2.5)	7 (3.0)	
40–49	59 (6.7)	47 (7.2)	12 (5.1)	
50–59	163 (18.4)	129 (19.9)	34 (14.3)	
60–69	258 (29.2)	191 (29.6)	67 (28.0)	
70–79	246 (27.7)	172 (26.5)	74 (31.0)	
≥80	120 (13.5)	80 (12.2)	40 (16.9)	
Sex				0.315
Male	421 (47.5)	314 (48.4)	107 (44.9)	
Female	465 (52.5)	334 (51.6)	131 (55.1)	
Insurance status				0.153
Medicare/Medicaid	509 (57.4)	362 (55.8)	147 (61.7)	
Private including HMO	346 (39.3)	263 (41.0)	83 (34.9)	
Self-pay/no-charge/other	29 (3.3)	21 (3.2)	8 (3.4)	
Missing	2	2	0	
Household income				0.225
Quartile 1	170 (19.6)	125 (19.7)	45 (19.3)	
Quartile 2	197 (22.5)	153 (24.0)	44 (18.6)	
Quartile 3	230 (26.6)	168 (26.6)	62 (26.6)	
Quartile 4	272 (31.3)	189 (29.8)	83 (35.5)	
Missing	17	13	4	
Smoking	306 (34.6)	211 (32.6)	95 (39.9)	**0.018**
Weekend admission	20 (2.2)	16 (2.5)	4 (1.7)	0.489
Hospital bed number				0.644
Small	54 (6.1)	39 (6.0)	15 (6.3)	
Medium	150 (17.1)	106 (16.5)	44 (18.7)	
Large	681 (76.9)	502 (77.5)	179 (75.0)	
Missing	1	1	0	
Hospital location/teaching status				**0.021**
Rural	8 (0.9)	4 (0.6)	4 (1.7)	
Urban nonteaching	74 (8.2)	51 (7.7)	23 (9.6)	
Urban teaching	803 (90.9)	592 (91.7)	211 (88.7)	
Missing	1	1	0	
Combined spleen removal	776 (87.5)	562 (86.7)	214 (89.9)	0.159
Emergent admission (missing = 1)	80 (9.1)	66 (10.2)	14 (6.0)	**0.016**
Comorbidities				
Coronary artery disease	140 (15.7)	93 (14.2)	47 (19.6)	**0.019**
Congestive heart failure	31 (3.4)	23 (3.4)	8 (3.4)	0.954
Persistent anemia	113 (12.6)	82 (12.5)	31 (12.9)	0.867
Diabetes	283 (31.8)	202 (31.0)	81 (34.0)	0.377
Hypertension	462 (52.0)	328 (50.5)	134 (56.3)	0.088
Cerebrovascular disease	14 (1.5)	10 (1.5)	4 (1.7)	0.782
Chronic pulmonary disease	120 (13.5)	87 (13.3)	33 (13.9)	0.823
Hyperlipidemia	343 (38.7)	242 (37.4)	101 (42.3)	0.174
Drug abuse	74 (8.4)	56 (8.7)	18 (7.6)	0.567
Severe Liver disease	17 (1.9)	14 (2.2)	3 (1.2)	0.306
Moderate or severe renal disease	43 (4.8)	33 (5.1)	10 (4.2)	0.552
Rheumatic disease	21 (2.4)	13 (2.0)	8 (3.4)	0.199
CCI				0.289
0–1	702 (79.4)	519 (80.3)	183 (76.9)	
2–3	138 (15.4)	97 (14.8)	41 (17.2)	
4–5	37 (4.2)	24 (3.7)	13 (5.4)	
6+	9 (1.0)	8 (1.2)	1 (0.4)	
Year of admission				0.556
2005–2014	359 (40.1)	267 (40.8)	92 (38.3)	
2015–2018	527 (59.9)	381 (59.2)	146 (61.7)	

CCI, Charlson Comorbidity Index; HMO, Health Maintenance Organization. Continuous variables are presented as mean ± SE. Categorical variables are presented as unweighted counts (weighted percentage). *p* < 0.05 is shown in bold.

**Table 2 cancers-16-01003-t002:** In-hospital outcomes following conventional vs. robot-assisted surgery.

In-Hospital Outcome	Total (*n* = 886)	Laparoscopic Surgery		*p*-Value
		Conventional (*n* = 648)	Robot-Assisted (*n* = 238)	
In-hospital mortality	9 (1.0)	9 (1.4)	0 (0.0)	-
Perioperative complication	212 (23.8)	168 (25.8)	44 (18.4)	**0.004**
AMI	1 (0.1)	1 (0.2)	0 (0.0)	-
CVA	6 (0.7)	4 (0.6)	2 (0.8)	0.708
VTE	33 (3.7)	29 (4.5)	4 (1.6)	**0.012**
Periprocedural shock, hypertension, or other cardiovascular complications	15 (1.7)	11 (1.7)	4 (1.6)	0.951
Pneumonia	27 (3.1)	22 (3.4)	5 (2.1)	0.256
Postprocedural pneumothorax	4 (0.5)	4 (0.6)	0 (0.0)	-
Postprocedural air leak	0 (0.0)	0 (0.0)	0 (0.0)	-
Acute respiratory failure	28 (3.2)	20 (3.1)	8 (3.4)	0.705
Pulmonary collapse (atelectasis)	64 (7.2)	50 (7.6)	14 (5.9)	0.297
Sepsis	32 (3.6)	25 (3.8)	7 (3.0)	0.435
Infection	32 (3.5)	26 (3.9)	6 (2.5)	0.182
Mechanical ventilation	15 (1.7)	11 (1.7)	4 (1.7)	0.989
Postoperative blood transfusion	61 (6.8)	53 (8.0)	8 (3.4)	**<0.001**
Other complications of the respiratory system	10 (1.1)	7 (1.1)	3 (1.3)	0.709
Perforations of organs or vessels	8 (0.9)	6 (1.0)	2 (0.8)	0.871
LOS, days ^a^	6.6 ± 0.2	6.8 ± 0.2	6.0 ± 0.2	**0.005**
Unfavorable discharge ^a^	57 (6.4)	42 (6.5)	15 (6.2)	0.861
Hospital cost, US dollars	99,874 ± 3506	95,402 ± 3602	112,036 ± 4567	**0.024**

AMI, acute myocardial infarction; CVA, cerebrovascular accident; VTE, venous thromboembolism; LOS, length of hospital stay. Continuous variables are presented as mean ± SE. Categorical variables are presented as unweighted counts (weighted percentages). *p* < 0.05 is shown in bold. ^a^ Excluded patients who died in the hospital.

**Table 3 cancers-16-01003-t003:** Associations between type of distal pancreatectomy (robot-assisted vs. conventional laparoscopic) and in-hospital mortality, unfavorable discharge, and perioperative complications.

Outcomes	Multivariable Analysis ^b^	
	aOR (95% CI)	*p*-Value
In-hospital mortality	NA	-
Unfavorable discharge ^a^	0.73 (0.43–1.24)	0.243
Perioperative complication, any	**0.61 (0.45–0.83)**	**0.002**
AMI	NA	-
CVA	1.09 (0.22–5.35)	0.912
VTE	**0.35 (0.14–0.85)**	**0.021**
Periprocedural shock, hypertension, or other cardiovascular complications	0.94 (0.31–2.84)	0.910
Pneumonia	0.56 (0.23–1.33)	0.184
Postprocedural pneumothorax	NA	-
Postprocedural air leak	NA	-
Acute respiratory failure	1.08 (0.61–1.89)	0.801
Pulmonary collapse (atelectasis)	0.70 (0.40–1.23)	0.219
Sepsis	0.70 (0.36–1.37)	0.298
Infection	0.62 (0.31–1.25)	0.182
Mechanical ventilation	0.92 (0.33–2.52)	0.864
Postoperative blood transfusion	**0.37 (0.23–0.61)**	**<0.001**
Other complications of the respiratory system	1.20 (0.50–2.86)	0.680
Perforations of organs or vessels	0.84 (0.17–4.23)	0.830

OR, odds ratio; aOR, adjusted OR; CI, confidence interval; AMI, acute myocardial infarction; CVA, cerebrovascular accident; VTE, venous thromboembolism; NA, no event occurred. *p* < 0.05 shown in bold. ^a^ Excluded patients who died in the hospital. ^b^ Adjusted for variables with a value of *p* < 0.05 in Table 1, including age (continuous), smoking, hospital location/teaching status, emergency admission, and coronary artery disease.

**Table 4 cancers-16-01003-t004:** Associations between type of distal pancreatectomy (robot-assisted vs. conventional laparoscopic), LOS, and total hospital costs.

Outcomes	Multivariable Analysis ^b^	
	aBeta (95% CI)	*p*-Value
LOS ^a^	**−0.76 (−1.43, −0.09)**	**0.026**
Total hospital costs	**18,284 (4369, 32,200)**	**0.010**

aBeta, adjusted β coefficient; CI, confidence interval; LOS, length of hospital stay. *p* < 0.05 shown in bold. ^a^ Excluded patients who died in the hospital. ^b^ Adjusted for variables with a value of *p* < 0.05 in Table 1, including age (continuous), smoking, hospital location/teaching status, emergency admission, and coronary artery disease.

## Data Availability

Data are contained within the article and Appendix A.

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
