# Peer review of "Short-Term Outcomes of Conventional Laparoscopic versus Robot-Assisted Distal Pancreatectomy for Malignancy: Evidence from US National Inpatient Sample, 2005–2018"

_cancers, 2024, doi:10.3390/cancers16051003_

Round 1

Reviewer 1 Report

Comments and Suggestions for Authors

This is a retrospective study of short-term outcomes of laparoscopic versus robot-assisted distal pancreatectomy for malignancy extracted from the US Nationwide Inpatient Sample (NIS) 2005-2018 database. This topic could be attractive for physicians performing these procedures. Some strengths of this study should be underlined: 1) A relatively large sample size (n=886); 2) Good revision of the literature. However, when a deep reading is performed several methodological concerns arise:

1. Patients are included during a large period (13 years!). The vast majority of patients were operated on by an open approach (around 258,000 cases) and no definitive criteria to select to laparoscopy or robot approach were shown. Therefore, some bias cannot be excluded (the best cases could have been chosen for minimally invasive surgery). Furthermore, the robotic approach is somehow newer than the laparoscopy approach and could have been used just in the last years of the period which again can be a source of bias., e.g. maybe the lower complication rate may be because better care has been provided to the patients the last years regardless the robotic approach. I would suggest to show in the paper when the patients for robotic approach was provided and how this may influence the final results. 

2. Minor concerns:

- In table 4, outcomes are expressed in aBeta coefficients instead the more conventional OR (odds ratio) which is always easier to be interpreted.

- In Figure 1, N=4,397 (bottom part) is shown. What does this figure refers to?

Author Response

Reviewer 1

This is a retrospective study of short-term outcomes of laparoscopic versus robot-assisted distal pancreatectomy for malignancy extracted from the US Nationwide Inpatient Sample (NIS) 2005-2018 database. This topic could be attractive for physicians performing these procedures. Some strengths of this study should be underlined: 1) A relatively large sample size (n=886); 2) Good revision of the literature. However, when a deep reading is performed several methodological concerns arise:

  1. Patients are included during a large period (13 years!). The vast majority of patients were operated on by an open approach (around 258,000 cases) and no definitive criteria to select to laparoscopy or robot approach were shown. Therefore, some bias cannot be excluded (the best cases could have been chosen for minimally invasive surgery). Furthermore, the robotic approach is somehow newer than the laparoscopy approach and could have been used just in the last years of the period which again can be a source of bias., e.g. maybe the lower complication rate may be because better care has been provided to the patients the last years regardless the robotic approach. I would suggest to show in the paper when the patients for robotic approach was provided and how this may influence the final results.

Author response:

We agree with the reviewer that surgical care improves over time, which could be a source of bias. To address this concern, we have added “year of admission” as a categorical variable in Table 1, categorizing patients into two periods: 2005-2014 and 2015-2018. It appears there was no significantly difference in patient distribution of year of admission between laparoscopic and robotic approach, thus further adjustment is not needed. Since this study was based on claim codes, definite criteria to select robotic rather than laparoscopic approach is unclear. We have added this into study limitations to remind the readers.

  1. Minor concerns:

- In table 4, outcomes are expressed in aBeta coefficients instead the more conventional OR (odds ratio) which is always easier to be interpreted.

Author response:

The presentation of outcomes in Table 4 as Beta coefficients is due to the nature of continuous variables. Our study did not further categorize Length of Stay (LOS) and cost into groups, which meant Odds Ratios (OR) could not be presented. Determining cut-offs for LOS and costs to convert them into binary variables could be considered arbitrary. As a result, we used Beta coefficients instead.

- In Figure 1, N=4,397 (bottom part) is shown. What does this figure refer to?

Author response:

In Figure 1, "N=4,397" means that the study sample of 886 patients can be statistically extrapolated to represent a larger population of 4,397 individuals in the entire United States. This extrapolation is achieved by applying the sample weights provided by the NIS dataset, allowing for broader generalizations beyond the specific patients included in the study. This is a standard approach as many published studies using the NIS, and has been already documented in Method and the first paragraph of Results (see 2.5. Statistical analysis and 3.1 Patient selection).

Reviewer 2 Report

Comments and Suggestions for Authors

I thank the editors for the opportunity to review the manuscript by Huang and colleagues about the short-term outcomes of laparoscopic versus robotic-assisted distal pancreactectomy using the US National Inpatient Sample. All in all, I congratulate the authors for their interesting and well-written analysis, that the readers may benefit greatly from. However, I still have a major issue I would like the authors to address in a revised version of the manuscript. I recommend to perform a propensity-score matched analysis to minimize confounding bias on the analysis. Because of the strong results of the presented study, I believe that selection bias favoring fitter patients for robotic surgery might be a possibility. Looking at the case numbers, a PSM analysis might be possible. Minor issue: why was 20 years of age chosen as the cutoff for the study?

Comments on the Quality of English Language

Minor grammatical issues, I recommend proof-reading by a native speaker.

Author Response

I thank the editors for the opportunity to review the manuscript by Huang and colleagues about the short-term outcomes of laparoscopic versus robotic-assisted distal pancreactectomy using the US National Inpatient Sample. All in all, I congratulate the authors for their interesting and well-written analysis, that the readers may benefit greatly from. However, I still have a major issue I would like the authors to address in a revised version of the manuscript.

I recommend to perform a propensity-score matched analysis to minimize confounding bias on the analysis. Because of the strong results of the presented study, I believe that selection bias favoring fitter patients for robotic surgery might be a possibility. Looking at the case numbers, a PSM analysis might be possible.

Author response:

Thank you for the insightful comments. Following consultation with our statistician, we decided to retain the current analysis. The main reason is that this study includes a total of 886 patients, with a ratio of approximately 1:3 between those undergoing robot-assisted and conventional surgery. Furthermore, most variables exhibit balance between the two groups. Considering these factors and to maintain a robust sample size, we advise against further PSM. If concerns regarding potential confounders persist, multivariable analysis has already been executed, with adjustments made for all significant factors identified in Table 1. However, if you still consider PSM beneficial for the analysis, we will revise our approach upon your recommendation.

Minor issue: why was 20 years of age chosen as the cutoff for the study?

Author response:

Our goal is to include the entire adult population, typically defined as individuals aged 20 years and above. By selecting this age threshold, we can accurately represent all adult patients across age groups, although it is rare for individuals under 50 years old to have pancreatic cancer, thereby highlighting how prevalence varies with age.

Comments on the Quality of English Language

Minor grammatical issues, I recommend proof-reading by a native speaker.

Author response:

We have thoroughly reviewed the manuscript and corrected all grammatical issues.

Reviewer 3 Report

Comments and Suggestions for Authors

very good article

Author Response

Author response:

Thank you so much for reviewing our work.

Reviewer 4 Report

Comments and Suggestions for Authors

Comments and Suggestions for Authors;

This study was aimed to compare short-term outcome after distal pancreatectomy for pancreatic cancer between conventional laparoscopic and robot-assisted distal pancreatectomy. This comparative study using large volume of patients of US Nationwide Inpatient Sample (NIS) might be very attractive one.

However there were some concerns in this analytic study as shown below.

1, Laparoscopic surgery has sometimes completed to undergone by conversion from laparoscopic to open surgery in the complicated cases. In this study, patients who underwent conventional laparoscopic surgery and robot-assisted surgery were completely undergone without any conversion such as from laparoscopy to open or from robot-assist to conventional laparoscopy.

These issues should be clarified in the material of patients. And also conversion ratesin two groups should be shown in this manuscript.

2,, Ratio of two laparoscopic surgeries was 648 versus 238 in this study. Only twenty-seven % of patients of distal pancreatectomy were performed by robot-assisted surgery. Usually robot-assisted surgery has been tried by fully experienced surgeons who are familiar with laparoscopic surgery. Therefore, robot-assisted surgery might be undergone by fully experienced hospital and surgeons.

Although hospital character about location such as rural or urban, and teaching or non-teaching were compared in this study, nothing was investigated about hospital volume and also surgeon`s volume.

These factors should be also analyzed in this study.

3, The main factor affecting surgical difficulty in distal pancreatectomy for pancreatic cancer has been recognized to be tumor factors such as tumor size, invading vasculatures and organs. However this study did not reveal any data about tumor staging factors as authors described about it. Therefore the result of this study could not be reliable one. Were there any data showing tumor stages such as combined resection of other organs including vasculatures in NIS data?

4, Operation time is strongly important for considering medical cost and hospital management. There was not shown about analytic results of operation time in this study even in Tables.

It should be clearly revealed about operation time of two groups of surgeries.

Author Response

Comments and Suggestions for Authors;

This study was aimed to compare short-term outcome after distal pancreatectomy for pancreatic cancer between conventional laparoscopic and robot-assisted distal pancreatectomy. This comparative study using large volume of patients of US Nationwide Inpatient Sample (NIS) might be very attractive one.

However, there were some concerns in this analytic study as shown below.

  1. Laparoscopic surgery has sometimes completed to undergone by conversion from laparoscopic to open surgery in the complicated cases. In this study, patients who underwent conventional laparoscopic surgery and robot-assisted surgery were completely undergone without any conversion such as from laparoscopy to open or from robot-assist to conventional laparoscopy.

These issues should be clarified in the material of patients. And also, conversion rates in two groups should be shown in this manuscript.

Author response:

We are fully aware and agree with the reviewer’s comment that conversion is crucial when assessing the outcomes between laparoscopic and robotic approaches. Regrettably, conversion cannot be well defined and captured using the claim codes. Accordingly, we acknowledge this issue as one of the major limitations of this study and warrant further investigation.

  1. Ratio of two laparoscopic surgeries was 648 versus 238 in this study. Only twenty-seven % of patients of distal pancreatectomy were performed by robot-assisted surgery. Usually robot-assisted surgery has been tried by fully experienced surgeons who are familiar with laparoscopic surgery. Therefore, robot-assisted surgery might be undergone by fully experienced hospital and surgeons.

Although hospital character about location such as rural or urban, and teaching or non-teaching were compared in this study, nothing was investigated about hospital volume and also surgeon`s volume.

These factors should be also analyzed in this study.

Author response:

We concur with the reviewer that hospital volume and surgeons' familiarity of the procedures are important factors. However, our analysis is constrained to the variables provided in the NIS dataset, which lacks data on surgeons' experience. This limitation has been duly noted in the limitations section of our study. While hospital volume could be inferred, our previous analytical experience has shown that hospital case volumes are strongly correlated with "Hospital location/teaching status." In this dataset, there is a noticeable overlap between high-volume hospitals and urban-teaching hospitals. Given the time constraints of this revision imposed by the journal, we did not undertake a reanalysis to incorporate hospital case volume. Should you deem the inclusion of hospital case volume crucial and beneficial to our analysis, we are willing to revise our approach and include hospital volume in our analysis based on your recommendation.

  1. The main factor affecting surgical difficulty in distal pancreatectomy for pancreatic cancer has been recognized to be tumor factors such as tumor size, invading vasculatures and organs. However, this study did not reveal any data about tumor staging factors as authors described about it. Therefore, the result of this study could not be reliable one. Were there any data showing tumor stages such as combined resection of other organs including vasculatures in NIS data?

Author response:

Unfortunately, the NIS database, from which our data were extracted, does not provide any detailed information on tumor staging, or data such as vasculatures. We have duly noted these limitations. Please check the updated Discussion section.

  1. Operation time is strongly important for considering medical cost and hospital management. There was not shown about analytic results of operation time in this study even in Tables.

It should be clearly revealed about operation time of two groups of surgeries.

Author response:

We appreciate the reviewer's emphasis on the importance of operation time. Due to the constraints of the claim-code based approach of this study, detailed information on operation time was not available for inclusion in the analysis. The NIS primarily focuses on inpatient data collected by the ICD codes, and certain variables, such as operation time, are not recorded. We have added this into study limitations.

Round 2

Reviewer 2 Report

Comments and Suggestions for Authors

I thank the authors for their revised manuscript and their comments on my review. Despite the large sample size, I still believe that a PSM analysis would be beneficial, especially because of the strong results. Therefore, I once again recommend to perform a PSM analysis.

Reviewer 4 Report

Comments and Suggestions for Authors

Cancers 2850137

Comments and Suggestions for authors;

Although authors revised the previous manuscript according to reviewers comments, most of required issues pointed out by reviewer could not properly reinvestigated and revised because of insufficient data NIS data available.

I think this study design using NIS data base might be in-appropriate for the aim to compare short-term outcomes of conventional laparoscopic and robotic assisted distal pancreatectomy for malignancy. This study was especially aimed to investigate and compare short-term outcomes after surgery for malignant lesions.  If this study would be aimed to compare these two surgical procedures of distal pancreatectomy only for ordinary benign lesions, it might make sense possibly.

Therefore, I regrettably think that the revised manuscript has to be rejected without any further revisions by authors.